# Effect of Punicalagin and Ellagic Acid on Human Fibroblasts In Vitro: A Preliminary Evaluation of Their Therapeutic Potential

**DOI:** 10.3390/nu16010023

**Published:** 2023-12-20

**Authors:** Rebeca Illescas-Montes, Manuel Rueda-Fernández, Anabel González-Acedo, Lucía Melguizo-Rodríguez, Enrique García-Recio, Javier Ramos-Torrecillas, Olga García-Martínez

**Affiliations:** 1Biomedical Group (BIO277), Department of Nursing, Faculty of Health Sciences, University of Granada, Avda. Ilustración 60, 18016 Granada, Spain; rebecaim@ugr.es (R.I.-M.); manuelrueda92@gmail.com (M.R.-F.); luciamr@ugr.es (L.M.-R.); ogm@ugr.es (O.G.-M.); 2Institute of Biosanitary Research, Ibs.Granada, C/Doctor Azpitarte 4, 18012 Granada, Spain; anabelglez@ugr.es (A.G.-A.); egr@ugr.es (E.G.-R.); 3Biomedical Group (BIO277), Department of Nursing, Faculty of Health Sciences of Melilla, University of Granada, C/Santander, 1, 52005 Melilla, Spain

**Keywords:** pomegranate, phenolic compounds, punicalagin, ellagic acid, fibroblasts

## Abstract

Background: Pomegranate is a fruit that contains various phenolic compounds, including punicalagin and ellagic acid, which have been attributed to anti-inflammatory, antioxidant, and anticarcinogenic properties, among others. Objective: To evaluate the effect of punicalagin and ellagic acid on the viability, migration, cell cycle, and antigenic profile of cultured human fibroblasts (CCD-1064Sk). MTT spectrophotometry was carried out to determine cell viability, cell culture inserts were used for migration trials, and flow cytometry was performed for antigenic profile and cell cycle analyses. Cells were treated with each phenolic compound for 24 h at doses of 10^−5^ to 10^−9^ M. Results: Cell viability was always significantly higher in treated versus control cells except for punicalagin at 10^−9^ M. Doses of punicalagin and ellagic acid in subsequent assays were 10^−6^ M or 10^−7^ M, which increased the cell migration capacity and upregulated fibronectin and α-actin expression without altering the cell cycle. Conclusions: These in vitro findings indicate that punicalagin and ellagic acid promote fibroblast functions that are involved in epithelial tissue healing.

## 1. Introduction

The pomegranate (*Punica granatum* L.) is a member of the *Punicaceae* family and is one of the oldest and best-known edible fruits. Over the years, the positive impact of pomegranate on human health has been demonstrated via various evidences, which has awakened a great interest in its consumption [1,2,3]. The pomegranate is a food with particular nutritional characteristics since its peel, seeds, and juice are rich in polyphenols. Thus, the juice is composed of approximately 85% water and 10% sugars in the form of glucose, sucrose, and fructose [4]. In addition, it contains a variety of organic acids such as ascorbic, citric, tartaric, and malic, which contribute to its acidic and refreshing taste, or amino acids such as glutamine, serine, aspartate, and alanine, which enhance the fruit as a food [5]. However, what really sets pomegranate apart is its phenolic compounds, known for their numerous health benefits [6]. These compounds include a number of polyphenols, such as condensed tannins, hydrolyzable tannins such as ellagitannins (punicalin and punicalagin) and gallotannins (digalloyl hexoside), as well as phenolic acids such as gallic and ellagic acid [5]. The red and purple tones of pomegranate are due to its anthocyanins, such as pelargonidin, cyanidin, and delphinidin. In addition, the fruit contains flavonoids such as catechin, epicatechin, luteolin, and quercetin [7,8]. The health benefits associated with these phenolic compounds are remarkable and cover a wide range of positive effects. Pomegranate has been recognized for its antioxidant properties, which help reduce oxidative stress in the body [9]. It is also recognized for its anti-inflammatory, antihypertensive, anticancer, antiulcer, antimicrobial, and hepatoprotective effects, making it a versatile and valuable dietary supplement [10,11,12,13]. Punicalagins are of particular biological interest due to their role as precursors of ellagic acid and other smaller bioactive phenolic compounds via hydrolysis [14]. This hydrolysis results from the action of tannase enzymes, leading to the formation of an intermediate compound. This intermediate compound subsequently undergoes spontaneous lactonization to produce ellagic acid. In turn, ellagic acid can be metabolized into urolithins, smaller molecules known for their potent antioxidant properties [15]. However, it should be noted that the absorption of orally ingested ellagitannins in the human gastrointestinal tract is still under debate. This lack of certainty derives from the fact that the bacterial species responsible for the transformation of punicalagins into ellagic acid are found mainly in the distal part of the gastrointestinal tract. This suggests that the bioavailability of ellagitannins may be limited, and the extent to which they may exert their health benefits in the human body is a subject of ongoing research [16].

Pomegranate polyphenols have demonstrated pharmacological and physiological benefits for various cell populations, including fibroblasts [17,18]. The latter is highly relevant in connective tissue, particularly for skin wound healing. Fibroblasts perform vital roles, such as collagen production, which is essential for wound healing. Additionally, fibroblasts participate in wound contraction and produce growth factors that encourage the restoration of impaired tissue [19]. Therefore, the consumption of pomegranate polyphenols could enhance fibroblast function and expedite the healing process of affected skin and connective tissue.

The skin healing process is a complex and intricate mechanism aimed at restoring damaged tissue. It comprises several stages, each of which plays a distinct role in the overall regeneration process. These stages overlap and are closely coordinated to ensure the successful healing of skin lesions. The main stages of skin healing are hemostasis, inflammation, proliferation, and maturation. During these stages, a variety of cell types, such as fibroblasts and endothelial cells, as well as numerous cytokines and growth factors, become integral components of the intricate signaling pathways that orchestrate the healing process [20,21]. In the hemostasis phase, different mechanisms are activated, leading to vasoconstriction and hemostasis, the main matrix element being fibrinogen. The activation of the coagulation cascade ends with the formation of a fibrin matrix that will allow the infiltration of different cell types to the site of injury [22,23]. Subsequently, the inflammatory phase occurs in which different proinflammatory mediators, such as TNF-α, IL-1β, IL-6, or IL-8, synthesized by neutrophils or monocytes, among others, are involved. These mechanisms make it possible to fight pathogens or debride the tissue by phagocytosis [24,25]. The proliferation phase develops days later, between 3 and 10 days after the wound. At this stage, granulation tissue formation and neovascularization occur, including re-epithelialization and immunomodulation. In this phase, fibroblasts play an important role since they have to proliferate and migrate in the wound bed, as well as begin their transformation process toward myofibroblasts that will allow wound contraction [26,27]. Finally, during the maturation and remodeling phase, neovascularization ceases, the production of granulation tissue decreases, and changes in ECM deposits, depositing more collagen type I and increasing the lysis of collagen type III, resulting in a mature scar [28].

As mentioned above, fibroblasts are essential in promoting the growth of granulation tissue in the wound bed, and they contribute to the synthesis of proteins, such as collagen and elastin, which are key components of the extracellular matrix (ECM). This biological structure is crucial for the repair process as it provides the necessary support for the remodeling of damaged tissue. Furthermore, as fibroblasts continue their activity, they mature and differentiate into myofibroblasts. Myofibroblasts are a specific category of cells within the healing process, characterized by their unique contractile properties, similar to those of smooth muscle cells. They express alpha-smooth muscle actin (α-actin), exhibit high levels of cytokines, and contribute to ECM synthesis [29]. Myofibroblasts are critical for wound closure as they use their contractile abilities to ensure complete and secure sealing of the wound. The transformative and dynamic role of fibroblasts, which evolve into myofibroblasts, demonstrates their importance in the overall skin healing process [30,31].

Pomegranate is considered a superfood that contains various healthy bioactive compounds in its peel, seed, or juice; however, few studies have investigated the effectiveness of these compounds in the repair of skin damage. Hence, the objective of this study was to determine the effect of punicalagin and ellagic acid, phenolic compounds present in pomegranates, on the viability, cell cycle, migration, and antigenic profile of cultured human fibroblasts.

## 2. Materials and Methods

### 2.1. Treatments and Cell Culture

The phenolic compounds used for treatments were punicalagin and ellagic acid (Sigma Chemical Co., St. Louis, MO, USA). Punicalagin was dissolved in methanol (5 mg/mL) and Milli-Q water (Millipore Corp, Bedford, MA, USA), whereas ellagic acid was diluted in 1 M NaOH (10 mg/mL) and Milli-Q water. Both were stored at −20 °C as the stock dilution, and were further studied at the concentrations detailed below.

Human dermal fibroblast cell line CCD-1064Sk was obtained from the American Type Cultures Collection (Ref: CRL-2076, ATCC, Manassas, VA, USA) by the cell bank of the Scientific Instrumentation Center of the University of Granada. Cells were cultured in Dulbecco’s Modified Eagle Medium (DMEM; Gibco Life Technologies Corporation, Staley Road, New York, NY, USA) with 100 IU/mL penicillin (ERN Laboratories, Barcelona, Spain), 50 μg/mL gentamicin (B. Braun Medical SA, Barcelona, Spain), 2.5 μg/mL amphotericin B (Sigma Chemical), 1% glutamine (Sigma Chemical), 2% HEPES (Sigma Chemical), and 10% of fetal bovine serum (FBS) (Gibco Life Technologies Corporation). Cultures were maintained at 37 °C in an atmosphere of 95% air humidity and 5% CO_2_.

Cells were separated from the culture flask using 0.05% trypsin and 0.02% ethylenediaminetetraacetic acid (EDTA) (Sigma Chemical). After detachment, cells were washed and suspended in a culture medium with 10% of FBS.

### 2.2. Cell Viability Assay

MTT spectrophotometry assay was used to quantify the reduction of yellow MTT (3-(4,5-dimethyl-2-thiazolyl)-2,5-diphenyl-2H-tetrazolium bromide) to an insoluble purple formazan product [1-(4,5-Dimethyl-2-thiazolyl)-3,5-diphenylformazan] by mitochondrial succinate dehydrogenase (Sigma Chemical). Briefly, cells were cultured at 1 × 10^4^ cells/mL per well in a 96-well plate (Falcon, Becton Dickinson Labware, NJ, USA) in a culture medium with 10% of FBS. Previously, fibroblasts were synchronized in DMEM supplemented with 2% of FBS during 24 h. After this period, the medium was then replaced with DMEM with 10% of FBS, along with one of the investigated phenolic compounds at concentrations of 10^−5^, 10^−6^, 10^−7^, 10^−8^, or 10^−9^ M (within the possible therapeutic dosage range). Cells cultured in DMEM medium with 10% FBS were used as controls, keeping them under identical conditions as the treated cells. At the 24 h mark, cells were washed with PBS and then cultured with phenol red-free DMEM to eliminate any potential interference from the dye, enhancing the utmost precision in our measurements. This medium, containing 0.5 mg/mL MTT (Sigma Chemical), was left to incubate for 4 h under standard cell culture conditions (temperature of 37 °C in an atmosphere of 95% air humidity and 5% CO_2_), resulting in the formation of a distinctive purple water-insoluble deposit, primarily composed of formazan crystals. Following this incubation period, the formazan crystals underwent dissolution by the addition of dimethyl sulfoxide (DMSO) (Sigma Chemical), and the absorbance of the resulting solution was measured at 570 nm employing a spectrophotometer (Sunrise TM, TECAN, Männedorf, Switzerland). Accordingly, subsequent assays exclusively utilized the treatment doses that demonstrated a significant increase in proliferation compared to control cells.

In addition, concentrations that increased cell viability by 50% (half maximal effective concentration, EC_50_) were calculated by applying a sigmoidal dose–response curve equation. The doses used in the proliferation experiments were selectively chosen based on their demonstrated effectiveness and subsequently applied in the remaining assays.

### 2.3. Cell Cycle

The evaluation of the phenolic compounds on the cell cycle was carried out by flow cytometry. Cells were cultured in 24-well plates for a period of 24 h, where they were exposed to the culture medium containing punicalagin or ellagic acid at doses of 10^−6^ or 10^−7^ M. After the incubation period, cells were gently detached from the culture plate by applying a solution of 0.05% trypsin and 0.02% EDTA (Sigma), followed by some washing steps with PBS. Afterward, we obtained a suspension of cells in phosphate-buffered saline (PBS), setting the stage for preparing cells for the subsequent cell cycle analysis. Therefore, the cell suspension was placed in 200 μL PBS combined with 2 mL 70% cold ethanol, vigorously mixed, and left for approximately 30 min in a cold environment (ice) before undergoing centrifugation. Following centrifugation, the pellet was reconstituted in 200 μL PBS. Subsequently, 100 μL RNase (1 mg/mL) and 100 μL propidium iodine were added to the cells. The mixture was then incubated at 37 °C for 30 min before undergoing analysis via flow cytometry, utilizing a 488 nm argon laser (Fast Vantage Becton Dickinson, Palo Alto, CA, USA). Results were expressed as the percentage of cells in distinct phases of the cell cycle (i.e., G0/G1, S, and G2/M).

### 2.4. Migration Assay

The migration assay was conducted by suspending human fibroblasts in a culture medium at a concentration of 4 × 10^5^ cells/mL. By using the chamber of the cell culture insert (ibidi, Munich, Germany), an amount of 70 μL of the cell suspension was added into each gap. Following a 24 h incubation period, the cell culture insert was removed, creating a defined cell-free space of 500 μm. The wells were then washed with PBS to remove any potential cellular remnants. Thereafter, punicalagin or ellagic acid was added at doses of 10^−6^ or 10^−7^ M, and the cells were then incubated under standard culture conditions. Images were captured at 0, 4, 8, 12, and 24 h after treatment using inverted phase contrast microscopy. Motic Images Plus 3.0 software (Motic, Hong Kong) was employed to quantify cell migration areas. The percentage of gap closure was determined using the following formula:Percentage Gap Closure (%) = (W_0_ − W_n_)/W_0_ × 100
where W_0_ represents the initial width of the space immediately following the culture insert assay, and W_n_ denotes the width at various measurement time points. This formula provided a quantitative measure of the reduction in the cell-free space over the specified time intervals, facilitating a comprehensive analysis of the impact of punicalagin and ellagic acid on human fibroblast migration.

### 2.5. Antigenic Profile Study

The fibroblasts’s antigenic profile involved a study of some array of monoclonal antibodies (mAbs). Specifically, the antibodies targeted essential markers such as anti-fibronectin and human anti-α-actin. These assays included using flow cytometry analysis for quantitative analysis and confocal microscopy for high-resolution imaging.

### 2.6. Flow Cytometry

After 24 h of culture with punicalagin or ellagic acid administered at doses of 10^−6^ or 10^−7^ M, the cells were separated from the 24-well plate with a solution composed of 0.05% trypsin and 0.02% EDTA. Subsequently, the cells were washed and suspended in PBS at a concentration of 5 × 10^4^ cells/mL. Cell permeabilization was performed with the Fix and Perm kit (Caltag Laboratories Inc., Burlingame, CA, USA). The cells were then labeled cells by direct staining with fibronectin and α-actin mAbs (Table 1). Next, 10 µL of the mAb was incubated with 100 µL of the initial cell suspension for 30 min at 4 °C in the absence of light. After a washing step and resuspension in 1 mL PBS, the cells were immediately analyzed by flow cytometry (Fast Vantage Becton Dickinson) with an argon laser at 488 nm. This analytical approach aimed to determine the percentage of fluorescent cells compared to the corresponding isotype controls, with counts based on the analysis of 2000–3000 cells.

### 2.7. Immunocytochemistry

The dermal fibroblasts were cultured in chamber slides (Sigma Chemical) at a concentration of 2 × 10^4^ cells/well. After a period of treatment of 24 h with punicalagin or ellagic acid at doses of 10^−6^ or 10^−7^ M. Cells were fixed in a cold methanol-acetone mixture (1:1) for 10 min and then rinsed with PBS. Following this, the cells were blocked with 10% FBS in PBS for 30 min and incubated for 2 h with the fibronectin or α-actin monoclonal antibody (mAb) at a 1:500 dilution, as outlined in Table 1. Following the removal of excess mAbs by washing, the cell nuclei were stained with DAPI (4,6-diamidino-2-phenylindole). Images of the stained cells were then captured using a Leica spectral confocal laser microscope (Leica Microsystems GmbH, Wetzlar, Germany).

### 2.8. Statistical Analysis

The mean and standard error of the mean (SEM) were calculated for all variables. Student’s t-test for two independent samples was employed for variables with a normal distribution, as determined by the Shapiro–Wilks test. For variables exhibiting a non-normal distribution, the Mann–Whitney U test was utilized. Data analyses were conducted using R software 4.3 (https://www.r-project.org), with statistical significance set at *p* ≤ 0.05. A minimum of three independent experiments were performed for each assay in order to promote a replicable approach and to ensure the robustness and reliability of our findings.

## 3. Results

### 3.1. Cell Viability Assay

As depicted in Figure 1, the fibroblast cell viability was significantly higher versus untreated controls in cultures treated with punicalagin (*p* < 0.038) or ellagic acid (*p* < 0.007) at all doses except for punicalagin at 10^−9^ M (*p* > 0.05). The EC_50_ value was 2.40 × 10^−5^ M for punicalagin and 69 × 10^−5^ M for ellagic acid.

### 3.2. Cell Cycle

Figure 2 depicts the cell cycle results after treatment for 24 h with the treatments. No significant changes in the percentage of cells in each cell cycle phase were observed in cells treated for 24 h with punicalagin or ellagic acid at 10^−6^ or 10^−7^ M compared with untreated cells. No DNA aneuploidy or signs of malignant transformation were observed.

### 3.3. Migration Assay

Migration assay results showed a significantly increased percentage gap closure in fibroblasts treated with punicalagin at 10^−7^ M or ellagic acid at 10^−6^ M versus untreated cells at all time points (*p* < 0.05). A significant increase in percentage gap closure was only observed at 24 h in the cells treated with 10^−6^ M punicalagin (*p* < 0.001) and at 4, 12, and 24 h in those treated with 10^−7^ M ellagic acid (*p* ≤ 0.003) (Figure 3).

### 3.4. Antigenic Profile

Flow cytometry results showed that treatment of the fibroblasts for 24 h with punicalagin or ellagic acid at doses of 10^−6^ of 10^−7^ M significantly increased their expression of fibronectin and α-actin mAbs in comparison to controls (Figure 4).

### 3.5. Immunocytochemistry

Flow cytometry findings were corroborated by the immunofluorescence study, which showed an increase in cells labeled with anti-α-actin (phycoerythrin) mAbs (red) and anti-fibronectin (fluorescein isothiocyanate) mAbs (green) (Figure 5) at 24 h of treatment with punicalagin or ellagic acid at doses of 10^−6^ or 10^−7^ M compared with controls.

## 4. Discussion

Wound healing involves various cell populations; however, fibroblasts hold particular significance. These cells are crucial in the formation of granulation tissue, which is a central component of the wound-healing process. The role of this cellular population is intricate, requiring a significant increase in the proliferation, migration, and differentiation of these cells. This augmentation is essential for fibroblasts to effectively carry out their diverse functions, including the deposition of extracellular matrix (ECM) components, secretion of growth factors, and facilitation of wound contraction [29]. All these processes contribute to the success of wound healing [20,32].

In this in vitro study, the viability and migration of cultured human fibroblasts and their expression of fibronectin and α-actin were significantly increased by treatment with punicalagin or ellagic acid, phenolic compounds present in pomegranate, at doses of 10^−6^ or 10^−7^ M. No cell cycle alterations were produced by either treatment. Similar effects have also been reported after in vitro treatment of human fibroblasts with other bioactive substances of natural origin, such as the phenolic compounds in extra virgin olive oil, where the authors demonstrated an increase in the proliferation and migration of human fibroblasts in culture after 24 h of treatment with hydroxytyrosol, tyrosol, or oleocanthal at doses similar to those used in our study [33].

According to the present study, punicalagin and ellagic acid are bioactive compounds that favor the viability of fibroblasts and increase their proliferative and migratory capacity. These results are in line with the findings of Bae et al. [34], who subjected human keratinocytes and dermal fibroblasts to UV-B radiation after treatment with ellagic acid at doses of 1 to 10 μM and observed a dose-dependent increase in cell viability at 24–48 h of culture, suggesting a potential skin antiaging effect. Likewise, Pacheco-Palencia et al. [18] demonstrated that 2 h of treatment with pomegranate polyphenolic extracts (37.5% ellagitannins and 2.7% ellagic acid) at doses of 0 to 60 mg/L produced a significant dose-dependent inhibition of UV-induced cell death in human skin fibroblasts (SKU-1064) exposed to UV-A and UV-B irradiation for 1 min. Previous in vitro and in vivo wound-healing assays have also found that treatment with pomegranate extracts reduced the surface area and healing time of wounds and increased cell proliferation and migration [35,36,37,38,39]. For instance, Yan H et al. [40] reported that polyphenols in pomegranate peel accelerated wound healing in diabetic mice by increasing fibroblast infiltration, collagen regeneration, and vascularization/epithelialization in the wound area. Recently, Marcelino et al. [41] found that exposure to bioactive compounds extracted from pomegranate leaves at concentrations of 50 to 400 μg/mL increased the viability of human fibroblasts by >70% and of keratinocytes by >50% at the highest concentrations. In a rat study, the application of hydrogels enriched with ellagic acid to wounds contaminated with *Staphylococcus aureus* and *Escherichia coli* reduced the wound area, inhibited infection, and promoted angiogenesis and collagen deposition [42].

The controlled and transient activation of myofibroblasts plays a crucial role in the restoration of tissue integrity. In this way, myofibroblasts upregulate the α-actin expression of stress fibers to generate a mechanically resistant wound, besides upregulating other cell markers such as vimentin and fibronectin [43]. Myofibroblasts are large, non-fusiform cells, with highly marked cytoskeleton, abundant endoplasmic reticulum, long and thin mitochondria, scant ribosomes, and no polysomes [44]. In the present study, the treatment of cultured human fibroblasts with 10^−6^ M punicalagin or 10^−7^ M ellagic acid significantly upregulated the expression of fibronectin and α-actin in comparison to untreated cells. These findings suggest that these phenolic compounds may not only increase fibroblast proliferation and migration but also favor the differentiation of fibroblasts to myofibroblasts, with positive effects on wound healing. In this sense, the phenolic compounds assayed could participate in signaling pathways that stimulate the expression of genes encoding proteins that play important roles in fibroblast proliferation, migration, and differentiation, such as FGF, VEGF, TGFβ1, PDGF, or COL-I, among others. In this context, previous studies have shown that plant-derived bioactive compounds such as luteolin, apigenin, ferulic, coumaric acid, or caffeic acid significantly increase the expression of these markers in the same cell population [45].

In addition to these effects observed in enhancing cell viability and wound contraction, the studied phenolic compounds may also actively participate in other processes related to wound healing, such as antioxidant or anti-inflammatory activity, or in other cancer-related processes. In this sense, pomegranate polyphenols have demonstrated antitumor capacities. Treatment with high concentrations of punicalagin or ellagic acid significantly reduced cell viability and increased apoptosis in U87MG glioma cells [46], human peripheral blood mononuclear cells, and HeLa cells and significantly decreased the viability, migration, and invasiveness of MCF-7 and MDA-MB-231 breast cancer cells [47]. It was also found that the treatment inhibited the permeability, proliferation, migration, and formation of VEGF-induced tubules in human umbilical vein endothelial cells [48] to reduce the proliferation of C6 glioma cells, promoting antiangiogenic processes [49] and inhibit the proliferation of Caco2 human colon adenocarcinoma cells [50]. The above authors applied higher concentrations of these polyphenols than in the present study, in which no cell cycle changes were produced.

These phytochemicals have also been found to exert anti-inflammatory activity [51]. In vitro treatment with ellagic acid, punicalagin, or pomegranate peel extract, alone or in combination with other compounds, was reported to induce an anti-inflammatory response, with significant reductions in TNF-α, IL-6, and IL-8 concentrations [52]. In an in vivo study of rats conducted by Deng et al., the treatment of skin burns with ellagic acid (at 50 or 100 mg/kg) was found to significantly reduce serum concentrations of IgA, IgG, IgM, TNF-α, IFN-γ, IL-1β, IL-6, and IL-10, while significantly increasing CD4+/CD8+ T cell and CD4+ CD25+ Treg cell counts, and decreasing the Foxp3 + T cell count. These changes suggest an anti-inflammatory effect and improvements in the immune system [53]. BenSaad et al. [51] attributed the anti-inflammatory effects of ellagic acid, gallic acid, and punicalagin on liposaccharide-treated RAW267.4 cells to the capacity of these polyphenols to inhibit nitric oxide, prostaglandin E2, and IL-6 production.

According to the present study, pomegranate polyphenols offer essential modulatory functions in wound healing. They not only promote cell proliferation, migration, and differentiation but can also assist in regulating the inflammatory response of lesioned tissue, reducing wound-healing time. They can even add value to the circular economy via their extraction from waste generated by the production of pomegranate juice.

## 5. Conclusions

The treatment of human fibroblasts in vitro with punicalagin or ellagic acid increases their viability and migration capacity and upregulates their expression of fibronectin and α-actin, which are crucially involved in wound healing, without producing cell cycle changes. Further in-depth investigation of the action mechanism of these compounds and their anti-inflammatory and antibacterial effects is warranted to determine their potential therapeutic role in the treatment of skin lesions.

## Figures and Tables

**Figure 1 nutrients-16-00023-f001:**
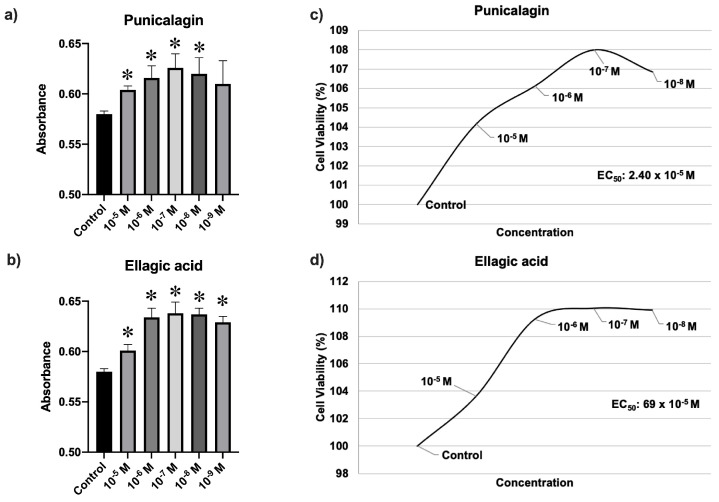
Effects of punicalagin (**a**) and ellagic acid (**b**) on the cell viability of fibroblasts at 24 h of treatment. EC_50_ value for each treatment (**c**,**d**). * *p* < 0.05. It is observed that treatment with punicalagin at doses of 10^−5^ to 10^−8^ M and ellagic acid at all concentrations significantly increase cell viability.

**Figure 2 nutrients-16-00023-f002:**
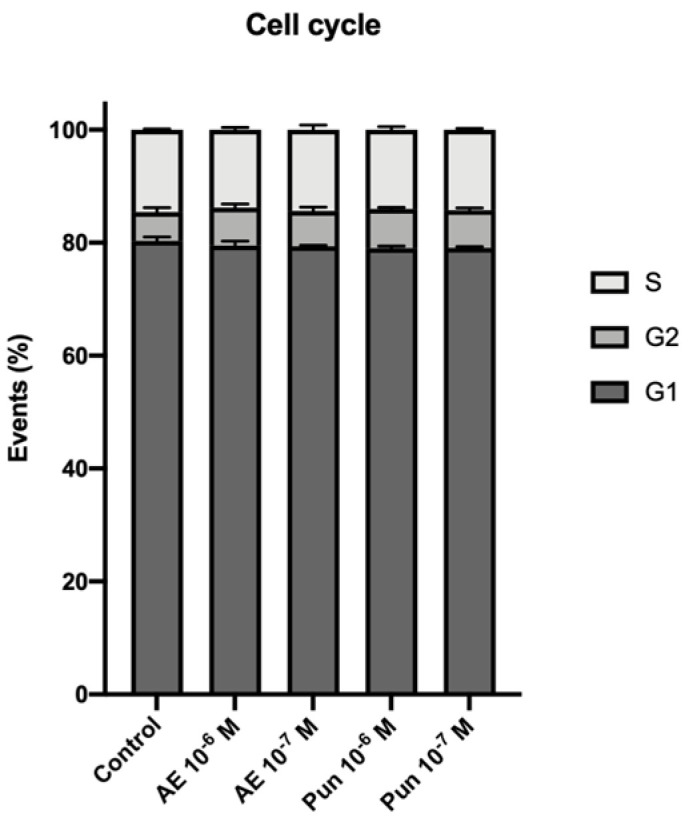
Percentages of cells in G0-G1, G2-M, and S phase at 24 h of treatment with punicalagin and ellagic acid as measured by flow cytometry. Significant alterations in the percentage of cells within each phase of the cell cycle are not observed. AE: ellagic acid; Pun: punicalagin.

**Figure 3 nutrients-16-00023-f003:**
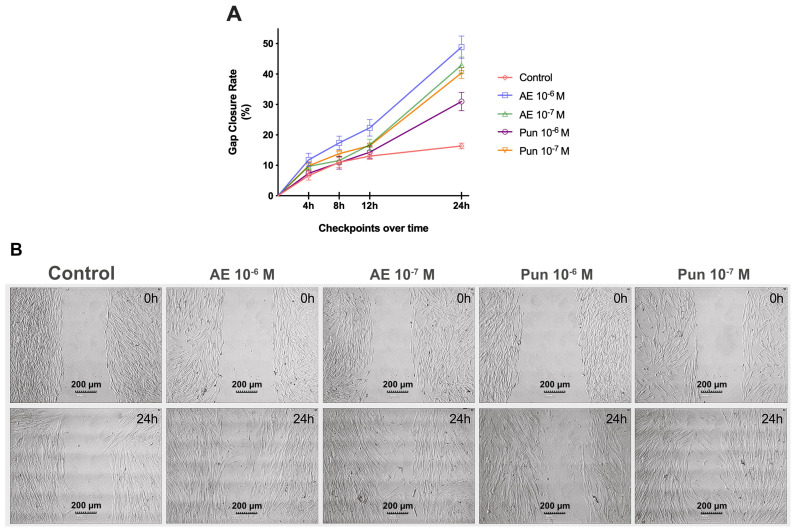
(**A**). Effect of punicalagin and ellagic acid on the migratory capacity of human fibroblasts. Punicalagin at 10^−7^ M and ellagic acid at 10^−6^ M significantly increased their capacity at 4 (*p* < 0.003 and *p* < 0.001), 8 (*p* < 0.005 and *p* < 0.001), 12 (*p* < 0.001 and *p* < 0.001), and 24 h (*p* < 0.001 and *p* < 0.001). Punicalagin at 10^−6^ M significantly increased it at 24 h (*p* < 0.001), and ellagic acid at 10^−6^ M significantly increased it at 4 (*p* < 0.003), 12 (*p* < 0.002), and 24 h (*p* < 0.001). AE: ellagic acid; Pun: punicalagin. (**B**). Bright-field microscope images of each treatment at 24 h. An observed reduction in the percentage of gap closure is evident across all treatments at varying doses compared to control cells. AE: ellagic acid; Pun: punicalagin.

**Figure 4 nutrients-16-00023-f004:**
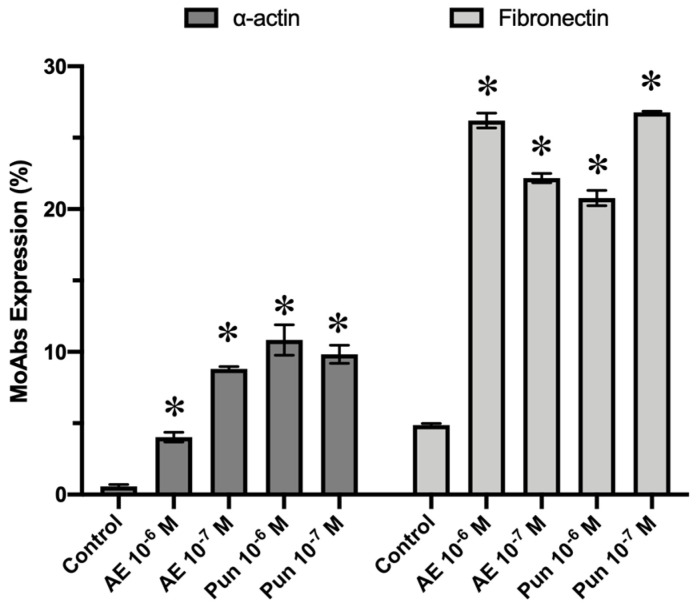
Expression of α-actin and fibronectin by cultured human fibroblasts at 24 h of treatment with punicalagin and ellagic acid. The percentage expression of α-actin and fibronectin was significantly higher in cells treated with different compounds at all selected doses. AE: ellagic acid; Pun: punicalagin. * *p* ≤ 0.05.

**Figure 5 nutrients-16-00023-f005:**
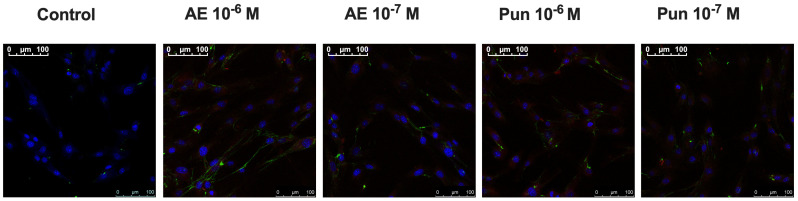
Immunostaining of human fibroblast cells where increased labeling of fibronectin-fluorescein (green) and α-actin-phycoerythrin (red) is observed in cells treated with punicalagin (Pun) and ellagic acid (AE) at the different doses tested at 24 h of treatment. The blue color in the nucleus shows DAPI staining.

**Table 1 nutrients-16-00023-t001:** Monoclonal antibodies (mAbs) used to study the antigenic phenotype of cultured human fibroblasts alongside the fluorochrome used to label the antibody and the supplier. FITC: Fluorescein-isothiocyanate. PE: Phycoerythrin.

mAb	Fluorochrome	Supplier
Control PE	PE	Caltag (Burlingame, CA, USA)
Control FITC	FITC	Caltag (Burlingame, CA, USA)
Anti-human Fibronectin-Fluorescein	FITC	R&D Systems (Minneapolis, MN, USA)
Anti-human α-Actin-PE	PE	R&D Systems (Minneapolis, MN, USA)

## Data Availability

Data are contained within the article.

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
