# Peer review of "Effect of Punicalagin and Ellagic Acid on Human Fibroblasts In Vitro: A Preliminary Evaluation of Their Therapeutic Potential"

_nutrients, 2023, doi:10.3390/nu16010023_

Round 1

Reviewer 1 Report

Comments and Suggestions for Authors

Illescas-Montes et al. have conducted research on the therapeutic potential of punicalagin and ellagic acid on cell viability, migration, etc, of cultured fibroblasts, highlighting potential benefits in wound healing. However, there are a few concerns the authors should address before the acceptance of this work.

Major:

1)    Does the in-vitro study suggest a direct application of punicalagin extracts on the epithelial tissues? Or if in animals/humans, it should be digested to achieve the therapeutic effect? What could be the bioavailability of punicalagin and ellagic acid in vivo, and what plasma concentration needs to be achieved to promote epithelial and fibroblast activities? The concentration at which the fibroblasts were affected in vitro, is it achievable in vivo? There is very limited information from solely an in-vitro study. Please use animal wound healing models with pharmacological studies as well.

2)    What mechanism of action would the authors propose? What are the potential pathways or effects on transcriptional or translational levels related to these two compounds?

3)    Cell staining in Figure 5 is not quantitative. Please use western blot to explore the potential alteration of protein expression in hypothesized pathways. Also, the staining images can at least be semi-quantified using fluorescence intensity measurement.

Minor:

1)    The experiment in 3.5/Figure 5 is immunocytochemistry, not immunohistochemistry.

2)    Figure 3B, the scratch is not visible. Please increase the image contrast.

Author Response

Reviewer #1:

Illescas-Montes et al. have conducted research on the therapeutic potential of punicalagin and ellagic acid on cell viability, migration, etc, of cultured fibroblasts, highlighting potential benefits in wound healing. However, there are a few concerns the authors should address before the acceptance of this work.

Major:

1) Does the in-vitro study suggest a direct application of punicalagin extracts on the epithelial tissues? Or if in animals/humans, it should be digested to achieve the therapeutic effect? What could be the bioavailability of punicalagin and ellagic acid in vivo, and what plasma concentration needs to be achieved to promote epithelial and fibroblast activities? The concentration at which the fibroblasts were affected in vitro, is it achievable in vivo? There is very limited information from solely an in-vitro study. Please use animal wound healing models with pharmacological studies as well.

 Response:

Thank you for your considerations, they are certainly accurate and are in line with our line of work. In this study we have developed different in vitro experiments in order to test the possible therapeutic usefulness of the phenolic compounds of pomegranate in wounds. To date, the scientific literature published with these compounds and their involvement in the healing process is limited, so we consider necessary the preliminary analysis of the effect of punicalagin and ellagic acid on human fibroblasts as a starting point for the further development of treatments in in vivo models.

In relation to bioavailability, in our work we propose the treatment with a series of concentrations as a screening in order to identify the most effective at the cellular level. In this sense, our results suggest that direct application of the selected concentrations on epithelial tissue would favor fibroblast proliferation, migration and differentiation. However, we cannot affirm that these concentrations would be the most effective in in vivo wound models, since these models involve other types of variables that are not present in in vitro assays.

2) What mechanism of action would the authors propose? What are the potential pathways or effects on transcriptional or translational levels related to these two compounds?

Response:

We consider interesting the question raised in relation to the possible mechanism of action of the compounds studied at the cellular level. Our hypothesis is centered on the idea that the phenolic compounds assayed could participate in signaling pathways that stimulate the expression of genes encoding proteins that play important roles in the fibroblast. We have added the following paragraph in the discussion section to clarify the possible mechanism of action of the compounds on the fibroblast:

Line 358: "In this sense, the phenolic compounds assayed could participate in signaling pathways that stimulate the expression of genes encoding proteins that play important roles in fibroblast proliferation, migration and differentiation such as FGF, VEGF, TGFβ1, PDGF or COL-I, among others. In this context, previous studies have shown that plant-derived bioactive compounds such as luteolin, apigenin, ferulic, coumaric acid or caffeic acid significantly increase the expression of these markers in the same cell population [45]."

3) Cell staining in Figure 5 is not quantitative. Please use western blot to explore the potential alteration of protein expression in hypothesized pathways. Also, the staining images can at least be semi-quantified using fluorescence intensity measurement.

Response:

Thank you once again for your contributions, which are undoubtedly useful to improve the work presented. In relation to the antigenic profile study, we have used flow cytometry analysis as a quantitative technique and confocal microscopy for high-resolution imaging as a qualitative study. As you can see in the results section, we include figure 4 where it is indicated "Expression of α-actin and fibronectin by cultured human fibroblasts at 24 h of treatment with punicalagin and ellagic acid". These data are qualitatively corroborated with the immunofluorescence study reflected in Figure 5.

Minor:

1) The experiment in 3.5/Figure 5 is immunocytochemistry, not immunohistochemistry.

Response:

Thank you very much for the correction, it was indeed a typographical error. We have changed the term "immunohistochemistry" to "immunocytochemistry" in the text.

2)Figure 3B, the scratch is not visible. Please increase the image contrast.

Response:

We have improved the quality of the image in figure 3B and we have divided the initial image into 2 independent figures that continue to have the same numbering. All these changes have been included in the text.

Reviewer 2 Report

Comments and Suggestions for Authors

Overall, this manuscript was an interesting read and highlighted the effects of two compounds with regards to their ability to expedite wound healing. My only major comment is that there is an insufficient number of cited references in the introduction and discussion. The rest of my comments are minor and are below.

General comments:

Introduction:

-line 33, sentence beginning with (SBW): 'The pomegranate', requires referencing, please amend.

-line 36, SBW: 'In addition', requires referencing, please amend.

-line 39, SBW: 'However,' requires referencing, please amend.

-line 40, SBW: 'These compounds', requires referencing, please amend.

-line 47, SBW: 'Pomegranate', requires referencing, please amend.

-line 50, SBW: 'Punicalagins', requires referencing, please amend.

-line 62, SBW: 'Furthermore', requires referencing, please amend. Also, delete 'Furthermore' and begin sentence with the next word.

-line 64, SBW: 'Fibroblasts', requires referencing, please amend.

-line 65, SBW: 'Additionally', requires referencing, please amend.

-line 94, SBW: 'Proteins', this sentence is incomplete. Please amend.

-line 99, SBW: 'They express', requires referencing, please amend.

Materials and Methods:

-line 140: please delete the word 'of'.

-line 144: what temperature did incubation occur? Please include details.

-line 161: what did you wash the cells with? Please include.

-line 165: please define what you mean by the term 'cold environment'? Please clarify.

Results:

-general comment: by convention, since tables and figures are stand alone, the results must be explained in the legend as it is not up to the reader to interpret the data. Please amend all accordingly.

-Figures 3b and 5: the images are too small to visualise and discern cells. Please enlarge. In addition, are there size bars (uM) in each image? If not, please include.

Discussion:

-line 287, SBW: 'This augmentation', requires referencing, please amend.

-line 294: please replace the word 'these' with 'either' and remove the letter 's' from the word 'treatments'.

-line 340, SBW: 'The treatment' the tense of the sentence is incorrect, please amend.

-line 346, SBW: 'These phytochemicals', requires referencing, please amend.

-line 349, SBW: 'In an' the tense of the sentence is incorrect, please amend.

Comments on the Quality of English Language

The manuscript requires minor editing as noted in comments above.

Author Response

Reviewer #2:

Overall, this manuscript was an interesting read and highlighted the effects of two compounds with regards to their ability to expedite wound healing. My only major comment is that there is an insufficient number of cited references in the introduction and discussion. The rest of my comments are minor and are below.

Response:

Thank you for your feedback and positive remarks on our manuscript. We appreciate your comment regarding the insufficient number of references in the paper. We acknowledge the importance of a well-supported theoretical framework and a comprehensive discussion, and therefore, we commit to incorporate additional relevant and current references. We are convinced that the suggested changes will allow us to strengthen and clarify our article.

General comments:

Introduction:

-line 33, sentence beginning with (SBW): 'The pomegranate', requires referencing, please amend.

Response:

We have now added the appropriate reference:

  1. Viuda-Martos, M.; Fernández-López, J.; Pérez-Álvarez, J. a. Pomegranate and Its Many Functional Components as Related to Human Health: A Review. Comprehensive Reviews in Food Science and Food Safety2010, 9, 635–654, doi:10.1111/j.1541-4337.2010.00131.x.

-line 36, SBW: 'In addition', requires referencing, please amend.

Response:

We have now added the appropriate reference:

  1. Mphahlele, R.R.; Fawole, O.A.; Mokwena, L.M.; Opara, U.L. Effect of Extraction Method on Chemical, Volatile Composition and Antioxidant Properties of Pomegranate Juice. South African Journal of Botany 2016, 103, 135–144, doi:10.1016/j.sajb.2015.09.015.

-line 39, SBW: 'However,' requires referencing, please amend.

Response:

We have now added the appropriate reference:

  1. Kandylis, P.; Kokkinomagoulos, E. Food Applications and Potential Health Benefits of Pomegranate and Its Derivatives. Foods 2020, 9, 122, doi:10.3390/foods9020122.

-line 40, SBW: 'These compounds', requires referencing, please amend.

Response:

We have now added the appropriate reference:

  1. Mphahlele, R.R.; Fawole, O.A.; Mokwena, L.M.; Opara, U.L. Effect of Extraction Method on Chemical, Volatile Composition and Antioxidant Properties of Pomegranate Juice. South African Journal of Botany 2016, 103, 135–144, doi:10.1016/j.sajb.2015.09.015.

-line 47, SBW: 'Pomegranate', requires referencing, please amend.

Response:

We have now added the appropriate reference:

  1. Lorzadeh, E.; Heidary, Z.; Mohammadi, M.; Nadjarzadeh, A.; Ramezani-Jolfaie, N.; Salehi-Abargouei, A. Does Pomegranate Consumption Improve Oxidative Stress? A Systematic Review and Meta-Analysis of Randomized Controlled Clinical Trials. Clinical Nutrition ESPEN 2022, 47, 117–127, doi:10.1016/j.clnesp.2021.11.017.

-line 50, SBW: ‘Punicalagins’, requires referencing, please amend.

Response:

We have now added the appropriate reference:

  1. Caballero, V.; Estévez, M.; Tomás-Barberán, F.A.; Morcuende, D.; Martín, I.; Delgado, J. Biodegradation of Punicalagin into Ellagic Acid by Selected Probiotic Bacteria: A Study of the Underlying Mechanisms by MS-Based Proteomics. J. Agric. Food Chem. 2022, 70, 16273–16285, doi:10.1021/acs.jafc.2c06585.

-line 62, SBW: 'Furthermore', requires referencing, please amend. Also, delete 'Furthermore' and begin sentence with the next word.

Response:

Thank you for your guidance. We have added the necessary reference as listed below and removed 'Furthermore' to start the sentence with the subsequent word as suggested.

  1. Guo, H.; Liu, C.; Tang, Q.; Li, D.; Wan, Y.; Li, J.-H.; Gao, X.-H.; Seeram, N.P.; Ma, H.; Chen, H.-D. Pomegranate (Punica Granatum) Extract and Its Polyphenols Reduce the Formation of Methylglyoxal-DNA Adducts and Protect Human Keratinocytes against Methylglyoxal-Induced Oxidative Stress. Journal of Functional Foods 2021, 83, 104564, doi:10.1016/j.jff.2021.104564.
  2. Pacheco-Palencia, L.A.; Noratto, G.; Hingorani, L.; Talcott, S.T.; Mertens-Talcott, S.U. Protective Effects of Standardized Pomegranate (Punica Granatum L.) Polyphenolic Extract in Ultraviolet-Irradiated Human Skin Fibroblasts. J. Agric. Food Chem. 2008, 56, 8434–8441, doi:10.1021/jf8005307

-line 64, SBW: 'Fibroblasts', requires referencing, please amend.

Response:

We have now added the appropriate reference:

  1. Atit, R.; Thulabandu, V.; Chen, D. Dermal Fibroblast in Cutaneous Development and Healing. Wiley Interdiscip Rev Dev Biol 2018, 7, 10.1002/wdev.307, doi:10.1002/wdev.307.

-line 65, SBW: 'Additionally', requires referencing, please amend.

Response:

Thank you for your observation. The reference required is the same as the one we previously added in response to your earlier comment (Reference 19). We have ensured this reference is appropriately cited in this instance as well.

-line 94, SBW: 'Proteins', this sentence is incomplete. Please amend.

Response:

Thank you for pointing out the incomplete sentence at line 94. We have rewritten the sentence for better clarity and completeness:

Line 94: “As mentioned above, fibroblasts are essential in promoting the growth of granulation tissue in the wound bed, and they contribute to the synthesis of proteins, such as collagen and elastin, which are key components of the extracellular matrix (ECM)”

-line 99, SBW: 'They express', requires referencing, please amend.

Response:

We have now added the appropriate reference:

  1. Phan, S.H. Biology of Fibroblasts and Myofibroblasts. Proc Am Thorac Soc 2008, 5, 334–337, doi:10.1513/pats.200708-146DR.

Materials and Methods:

-line 140: please delete the word 'of'.

Response:

We have removed this word as suggested.

-line 144: what temperature did incubation occur? Please include details.

Response:

Thank you for your enquiry about the incubation conditions of line 144. We have clarified the incubation conditions:

“This medium, containing 0.5 mg/ml MTT (Sigma Chemical) was left to incubate for 4 h under standard cell culture conditions (temperature of 37 °C in an atmosphere of 95% air humidity and 5% CO2)…”

-line 161: what did you wash the cells with? Please include.

Response:

We have amended the manuscript to include the information that the washing steps were conducted using Phosphate-Buffered Saline (PBS). We have added in the text:

“…followed by some washing steps with PBS.”

-line 165: please define what you mean by the term 'cold environment'? Please clarify.

Response:

Thank you for your comment. To provide specificity, we have indicated that this refers to the use of ice, thereby clarifying that the procedures were conducted in an environment chilled with ice. We have added in the text:

“Therefore, the cell suspension was placed in 200 μL PBS combined with 2 ml 70% cold ethanol, vigorously mixed, and left for approximately 30 min in a cold environment (ice) before undergoing centrifugation.”

Results:

-general comment: by convention, since tables and figures are stand alone, the results must be explained in the legend as it is not up to the reader to interpret the data. Please amend all accordingly.

Response:

We have revised all the legends to include the main results and key interpretations. This ensures that each table and figure is comprehensible and stands alone, without requiring the reader to interpret the data from the text.

-Figures 3b and 5: the images are too small to visualise and discern cells. Please enlarge. In addition, are there size bars (uM) in each image? If not, please include.

Response:

Thank you for your feedback regarding Figures 3b and 5. We have modified the figures to enlarge the images for better visibility and clarity of them. Additionally, we have separated the figure 3A and 3B, and we have changed size bars, measured in micrometers (µm), to make them visible in each image to provide a clear scale reference.

Discussion:

-line 287, SBW: 'This augmentation', requires referencing, please amend.

Response:

We have now added the appropriate reference:

  1. Phan, S.H. Biology of Fibroblasts and Myofibroblasts. Proc Am Thorac Soc 2008, 5, 334–337, doi:10.1513/pats.200708-146DR.

-line 294: please replace the word 'these' with 'either' and remove the letter 's' from the word 'treatments'.

Response:

Thank you for your suggestion. We have replaced 'these' with 'either' and removed the 's' from 'treatments' as advised.

-line 340, SBW: 'The treatment' the tense of the sentence is incorrect, please amend.

Response:

Thank you for pointing out the tense issue. We have rewritten the sentence to ensure the correct tense is used.

-line 346, SBW: ‘These phytochemicals’, requires referencing, please amend.

Response:

We have now added the appropriate reference:

  1. BenSaad, L.A.; Kim, K.H.; Quah, C.C.; Kim, W.R.; Shahimi, M. Anti-Inflammatory Potential of Ellagic Acid, Gallic Acid and Punicalagin A&B Isolated from Punica Granatum. BMC Complement Altern Med 2017, 17, 47, doi:10.1186/s12906-017-1555-0.

-line 349, SBW: 'In an' the tense of the sentence is incorrect, please amend.

Response:

We have rewritten the text to ensure its coherence and correctness in accordance with your suggestions.

The manuscript requires minor editing as noted in comments above.

Response:

Thank you for your review and constructive comments. We hope that we have addressed the minor editing requirements noted to improve and clarify the manuscript.

Round 2

Reviewer 1 Report

Comments and Suggestions for Authors

The authors have made acceptable response to my previous concerns.